# Haptics-based Curiosity for Sparse-Reward Tasks

**Sai Rajeswar**[1,2,3,*] **Cyril Ibrahim**[3,*] **Nitin Surya**[3],
**Florian Golemo**[1,2,3]**, David Vazquez**[3]**, Aaron Courville**[1,2,4]**, Pedro O. Pinheiro**[3]
[3]Montréal Institute for Learning Algorithms, [2]Université de Montréal,
[3]Element AI, Montréal, [4]CIFAR Fellow

**Abstract:** Robots in many real-world settings have access to force/torque sensors in their gripper and tactile sensing is often necessary for tasks that involve contact-rich motion. In this work, we leverage surprise from mismatches in haptics feedback to guide exploration in hard sparse-reward reinforcement learning tasks. Our approach, Haptics-based Curiosity (HaC), learns what visible objects interactions are supposed to "feel" like. We encourage exploration by rewarding interactions where the expectation and the experience do not match. We test our approach on a range of haptics-intensive robot arm tasks (e.g. pushing objects, opening doors), which we also release as part of this work. Across multiple experiments in a simulated setting, we demonstrate that our method is able to learn these difficult tasks through sparse reward and curiosity alone. We compare our cross-modal approach to single-modality (haptics- or vision-only) approaches as well as other curiosity-based methods and find that our method performs better and is more sample-efficient.

## 1 Introduction

Most successes in reinforcement learning (RL) come from games [1, 2] or scenarios where the reward is strongly shaped [3, 4]. In the former, the environment is often fully observable, and the reward is dense and well-defined. In the latter, a large amount of work is required to design useful reward functions. While it may be possible to hand-craft dense reward signals for many real-world tasks, we believe that it is a worthwhile endeavor to investigate learning methods that do not require dense rewards.

Closely related to the sparse rewards problem is the issue of exploration. One reason that traditional RL agents struggle with sparse-reward problems is a lack of exploration. An agent may not obtain useful rewards without an intuitive exploration strategy when rewards are sparse. Exploration based on intrinsic curiosity comes naturally to many animals and infants (who start crawling and exploring the environment at around 9 months [5] and oftentimes even before they can crawl by using their hands and mouth to touch and probe objects). Touch is a local experience and encodes accurate geometrical information while handling objects. Experimental studies in infants has suggested that tactile and visual sensory modalities play a central role in systematic learning of tasks related to object understanding, interaction and manipulation [6, 7].

Ideally, we would like our RL agents to explore the environment in an analogous self-guided fashion to learn the dynamics and object properties, and use this knowledge to solve downstream tasks. Just as how humans utilize different sensory modalities to explore and understand the world around them, exploration in robots should be more embodied and related to a combination of vision and touch and potentially other sensor modalities. We believe that building autonomous agents that are self-driven and seek to explore via multi-modal interaction are crucial to address key issues in developmental robotics.

Recent works in RL have focused on a curiosity-driven exploration through prediction-based surprise [8, 9, 10]. In this formulation, a forward dynamics model predicts the future, and if its prediction is incorrect when compared to the real future, the agent is surprised and is thus rewarded. This encourages the agent to look for novel states while improving its visual forward model in return. However, this formulation can be practically challenging to optimize since there are many

---

*Equal contribution. Correspondence to: Sai Rajeswar<sai.rajeswar.mudumba@umontreal.ca>

5th Conference on Robot Learning (CoRL 2021), London, UK.

states that are visually dissimilar but practically irrelevant (e.g. for a pushing task, moving a robotic end-effector without touching the object creates visual novelty but contributes little to task-related knowledge). One way to constrain this search space over curious behaviors is by involving another modality like haptics.

In this work, we demonstrate that a self-guided cross-modal exploration policy can help solve sparse-reward downstream tasks that existing methods without this curiosity struggle to solve. Our method uses cross-modal consistency (mismatch between visual and haptic signal) to guide this exploration. To use self-play knowledge in downstream tasks, we relabel past experiences, providing a dense reward signal that allows modern off-policy RL methods to solve the tasks. While there are many existing methods that use artificial curiosity/intrinsic motivation, the majority of these methods either rely on strong domain knowledge (e.g. labels of state dimensions in Forestier et al. [4], a goal-picking strategy in Andrychowicz et al. [11]) or are prone to get stuck in local optima when a single meaningless stimulus creates enough surprise to capture the attention of the agent (e.g. noisy-TV experiment from [12]). Other approaches depend on unrealistic assumptions and goal conditioning [11]. Our method presents a novel approach in the family of prediction-based models [8, 13, 9] and yields better performance on a wide range of robotic manipulation tasks than purely vision-based and haptics-based approaches [8, 9]. The tasks are chosen with careful consideration—they comprise of preliminary robotic manipulations such as grasping, pushing, and pulling. In this work, we present the following contributions:

- A new curiosity method[2] to help solve sparse-reward tasks that use cross-modal consistency (predicting one modality from another) to guide exploration. We implement it in this work as vision and touch modalities, but the formulation of our method does not require any knowledge about the underlying modalities and can thus be applied to other settings.

- We create and maintain a manipulation benchmark of simulated tasks, *MiniTouch*, inspired by Chen et al. [14], Andrychowicz et al. [11], where the robotic arm is equipped with a force/torque sensor. This allows evaluation of models' performance on different manipulation tasks that can leverage cross-modal learning.

- We validate the performance of our method on MiniTouch environment comprising of four downstream tasks. We compare purely vision-based curiosity approaches and standard off-policy RL algorithms. Our method improves both performance and sample efficiency.

## 2   Related Work

**Intrinsic Motivation**   Intrinsic motivation is an inherent spontaneous tendency to be curious or to seek something novel in order to further enhance one's skill and knowledge [15, 16, 17]. This principle is shown to work well even in the absence of a well-defined goal or objective. In reinforcement learning, intrinsic motivation has been a well-researched topic [18, 19, 20, 21, 22, 23]. An intuitive way to perform intrinsic motivation is through the use of "novelty discovery". For example, incentivize the RL agent to visit unusual states or states with substantial information gain [24]. In its simplest form, this can be achieved with up-weighting mechanisms such as state visitation counts [25]. Count-based methods have also been extended to high-dimensional state spaces [26, 12, 27]. Alternative forms of intrinsic motivation include disagreement [28], empowerment [29, 30].

Exploratory intrinsic motivation can also be achieved through "curiosity" [18, 31]. In this setting, an agent is encouraged to visit states with high predictive errors [10, 8, 9] by training a forward dynamics model that predicts the future state given the current state and action. Instead of making predictions in the raw visual space, Pathak et al. [9] mapped images to a feature space where relevant information is represented via an inverse dynamics model. Burda et al. [8] demonstrate that random features are sufficient for many popular RL game benchmarks. This approach may work well with tasks that require navigation to find a reward because each unseen position of the agent in the world leads to high intrinsic reward when unseen. However, in the case of manipulative tasks, we are less interested in the robot visiting all the possible states and more interested in states where the robot interacts with other objects. In this work, we leverage multimodal inputs that encourage the agent to find novel combinations of visual and force/torque modalities.

---

[2]Project website: https://fgolemo.github.io/haptics-based-curiosity/

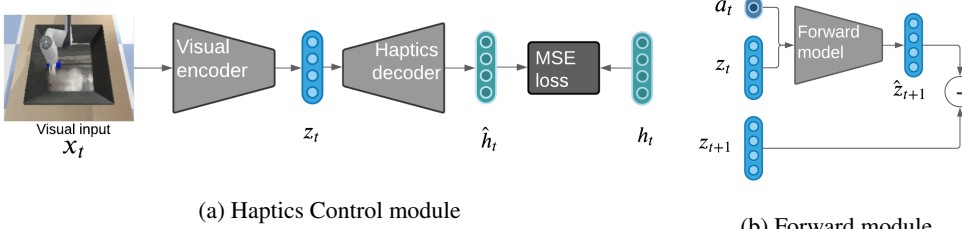

(a) Haptics Control module

(b) Forward module

Figure 1: **Haptics-based Curiosity Model.** (a) The input image $x_t$ at time $t$ is transformed into a 256-dimensional feature vector $z_t = enc(x_t)$ using a CNN encoder. The haptics decoder network predicts corresponding force/torque vector $\hat{h}_t = dec(z_t)$. The $L2$ norm between predicted haptics $\hat{h}_t$ and observed haptics $h_t$ is used as exploration reward. (b) To stabilize training, an additional network is used to predict the forward dynamics, and the difference between predicted next latent state $\hat{z}_{t+1}$ and actual next latent state $z_{t+1}$ is used as weighted additional term in the reward.

**Self-Supervised Learning via Cross-modality**  Exploiting multimodality to learn unsupervised representations dates back to at least 1993 [32]. Multimodal signals are naturally suitable for self-supervised learning, as information from one modality can be used to supervise learning for another modality [33, 34]. Different modalities typically carry different information, e.g., visual and touch sensory modalities emerge concurrently and often in an interrelated manner during contact-rich manipulation tasks [35]. Specifically, force/torque motor signal has always been a major component in the literature of perception and control [36, 37, 38].

The most common ways to leverage multimodal signals to learn representations are through vision and language [39, 40]. Gao et al. [41], Li et al. [42] demonstrated a unified approach to learning representations for prediction tasks using visual and touch data. Chen et al. [43] learn world-models from multimodal data via a shared latent space. In robotics and interactive settings, the use of modalities such as tactile sensing [44, 45, 46] is increasingly popular for grasping, manipulation and other externally-specified tasks [47, 48]. Lee et al. [49] showed the effectiveness of self-supervised training of tactile and visual representations by demonstrating its use on a peg insertion task.

While the mentioned approaches have used multiple sensory modalities for learning better representations, in this work we demonstrate its utility for allowing agents to explore. Similar to ours, Dean et al. [50] use multimodal sensory association (i.e. audio and visual) to compute the intrinsic reward. Their curiosity-based formulation allows an agent to efficiently explore the environment in settings where audio and visual signals are governed by the same physical processes. In addition to the different nature of sensory signals, they use a discriminator that determines whether an observed multimodal pair is novel. This might not work in our case as touch is a more sparse signal and using a discriminator could lead to ambiguous outcomes.

## 3 Proposed Approach

The goal of our method, Haptics-Based Curiosity (HaC), is to encourage the agent to interact with objects. HaC provides a reward signal for an RL agent to explore the state space of a task that involves interacting with objects. The exploration phase is independent of the downstream task, i.e., relying solely on visual and force/torque signals, without a reward signal from the downstream task.

Similar to how people spend more time exploring stimuli that are more incongruous [7], HaC guides the agent to focus its experience on different novel cross-modal associations. We augment this intrinsic objective with a future visual state prediction objective similar to the one in Pathak et al. [9] to avoid getting stuck in undesired inactive configurations. Note that we sometimes refer to the future state prediction objective in the text as forward dynamics objective. In this work, we focus on the cross-modality between vision and touch, but the same idea could be applied to other pairs of sensory domains, such as vision and sound, or touch and acoustics.

### 3.1 Problem Formulation

The learning problem is formalized as a Markov decision process (MDP) defined by a tuple $\{S, A, T, R, \rho, \gamma\}$ of states, actions, transition probability, reward, initial state distribution, and

discount factor. The goal is to find the optimal policy that maximizes the discounted sum of rewards, $\pi^* = \mathbb{E}_\pi[\sum_t^\infty \gamma^t r(\mathbf{s}_t, \mathbf{a}_t)]$. In our case, each state $s_t$ in the trajectory is composed of both visual and corresponding touch features as detailed in the following section. We use soft actor-critic policy gradients (SAC) [51] to train our policies, but in principle, our proposed approach is algorithm-agnostic. The policy $\pi$ is evaluated with an estimation of the soft Q-value:

$$Q(\mathbf{s}_t, \mathbf{a}_t) \triangleq r(\mathbf{s}_t, \mathbf{a}_t) + \gamma \mathbb{E}_{\mathbf{s}_{t+1} \sim p}[V(\mathbf{s}_{t+1})] \,, \tag{1}$$

where $V(\mathbf{s}_t) = \mathbb{E}_{\mathbf{a}_t \sim \pi}[Q(\mathbf{s}_t, \mathbf{a}_t) - \log \pi(\mathbf{a}_t|\mathbf{s}_t)]$ is the soft value function.

## 3.2 Haptics-based Prediction

Our core prediction framework consists of two modules: (i) a *haptics control* module, which learns to predict expected haptic signal from the visual input, and (ii) a *forward dynamics* model, which predicts the next latent state from the current latent state and the current action (see Figure 1). Let the state of the environment $\mathbf{s}_t = (z_t, h_t)$ at time $t$ be composed of a visual feature $z_t$ (the encoded visual input) and a haptic signal $h_t$. The touch prediction model consists of a convolutional encoder $z_t = enc(x_t)$ that transforms the image $x_t$ into a latent representation $z_t$ and a fully-connected decoder $\hat{h}_t = dec(z_t)$ that transforms the latent into a predicted haptic signal $\hat{h}_t$. The encoder-decoder is trained with an L2 reconstruction loss, i.e. for every image $x_t$ and force/torque sensor $h_t$:

$$L_{haptics} = \left\| \hat{h}_t - h_t \right\|_2 \,. \tag{2}$$

A high prediction error on a given image indicates that the agent has had few interactions like this. Therefore, to harness this "surprise" to guide exploration, we define the intrinsic reward at time $t$ during exploration to be proportional to this reconstruction loss. This essentially allows the agent policy to visit under-explored configurations of the state space by encouraging interactions where the system does not know what the target object "feels" like. In addition to efficient exploration, being aware of such incongruity via touch prediction aids learning local regularities in the visual input. This in turn could assist better generalization to unseen states. An overall pipeline of the framework is shown in Figure 1a.

## 3.3 Regularization Through Forward Dynamics Model

We found empirically (and we demonstrate in the experiments section below) that the surprise stemming from haptic novelty was not enough to cause object-centric interaction. We postulate that by incorporating visual surprise (i.e. the mismatch between predicted forward dynamics and observed dynamics) [9], we can create an agent that seeks out visual novelty as well as haptic one and thus leads to better state space coverage. To this end, our model is augmented with a forward dynamics model (see Figure-1b) $\hat{z}_{t+1} = fdm(z_t, a_t)$ that learns to map the latent state $z_t$ (obtained from the visual encoder $enc$) and action $a_t$ at time $t$ to the predicted latent state $\hat{z}_{t+1}$ at the next timestep. This model is also trained with L2 loss:

$$L_{fdm} = \|\hat{z}_{t+1} - z_{t+1}\|_2 = \|fdm(enc(x_t), a_t) - enc(x_{t+1})\|_2 \,. \tag{3}$$

The intrinsic reward is defined as the convex combination of the cross-modal prediction loss (Eq. 2) and the forward dynamics model loss (Eq. 3):

$$r_t = (1 - \lambda) \cdot L_{haptics} + \lambda \cdot L_{fdm} \,, \tag{4}$$

where $\lambda \in [0, 1]$ is a balancing factor. The effect of the factor $\lambda$ on overall performance is studied in the ablation experiments described in Section 5.1.

## 3.4 Training

Learning is divided into two stages: (i) an *exploratory* step, where the agent performs free exploration following HaC, and (ii) an *adaptation* step, where the agent is tasked to solve a downstream problem, given a sparse reward.

During the exploratory step, each trajectory consists of pairs of image and force/torque features, $(z_1, h_1), (z_2, h_2),..., (z_n, h_n)$. These trajectories are used for two purposes: (i) updating the parameters of the prediction model to help shape the representations and (ii) updating the exploration policy

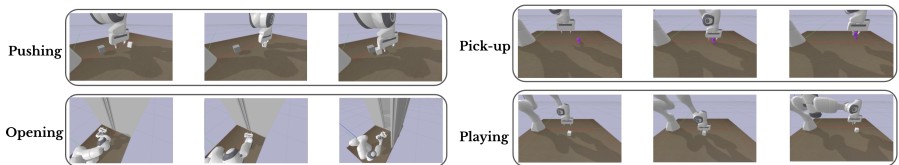

Figure 2: **'MiniTouch' benchmark tasks** include opening a door, pushing an object to a target, grasping and lifting an object and a toy task in which object interactions are counted.

based on the intrinsic reward $r_t$. Note that we use the touch features only to craft the intrinsic rewards and the input to RL agent consists of visual features alone to be comparable to the baselines. For vision-based curiosity models, Burda et al. [8] observed that encoding visual features via a random network constitute a simple and effective strategy compared to learned features. In Section 5.1, we investigate the performance of our model in both scenarios, i.e., when the features are learned vs random. The overall optimization problem at this step consists of the policy learning (driven by intrinsic reward), the touch reconstruction loss (Eq. 2), and the forward dynamics loss (Eq. 3). During the downstream adaptation step, the parameters of the policy network, the Q network and the replay buffer are retained from the exploratory phase. The objective of the down-stream task is computed as:

$$\min_\theta \left[ -\mathbb{E}_\pi \left[ \sum_t r_t^e \right] \right] \; , \tag{5}$$

where $r_t^e$ in this phase is task-specific external sparse reward. In both steps, the objectives are optimized with Adam [52].

**Implementation Details**: The encoder is a four-layered strided CNN followed by a fully connected network. We use LeakyReLU [53] as non-linear activation in all the layers. The decoder network is a two-layered MLP that maps 256-dimensional visual features to touch vectors. For a more detailed description of the networks (SAC policy network and HaC networks, including the forward model) and hyper-parameters, please refer to the supplementary material Section 7.3.

## 4  Experimental Setting

Our experiments focus on a tabletop robot manipulation from raw image observations and raw force/torque sensory values, which we refer to as "touch vector". For our experiments, we use a 7-DoF Franka Emika Panda arm with a two-finger parallel gripper. Each of the fingers is equipped with a simulated force/torque sensor that measures the joint reaction force applied to it. We utilize PyBullet [54] to simulate the robot arm and haptic sensor[1].

### 4.1  MiniTouch Benchmark

Our proposed benchmark, MiniTouch, consists of four manipulation tasks: Playing, Pushing, Opening, Pick-up. Each of the tasks along with corresponding actions, observations, and rewards is described in detail in the Supplementary Material Section 7.1 and further illustrated in Figure 2. MiniTouch is an active repository[2] and we expect to update the benchmark with new tasks and datasets. The tasks are inspired by Yu et al. [55] but are *not* based on a proprietary simulator, feature an arm that we have access to for real-world experimentation (in follow-up work), and where the arm is equipped with a haptic sensor.

### 4.2  Baseline Comparisons and Metrics

For the task evaluation, we study two versions of our model: (i) HaC-Pure considering the touch vector reconstruction intrinsic reward alone, and (ii) HaC, considering the full intrinsic reward (Eq. 4). We compare with several well-known intrinsic exploration baselines based on visual features:

- **SAC**: The unmodified Soft Actor-Critic algorithm from Haarnoja et al. [51].
- **ICM**: SAC augmented with the state-of-the-art visual curiosity approach Intrinsic Curiosity Module (ICM) [12], which uses a visual prediction model to guide exploration.

---

[1]Based on code from the official Franka Emika repo https://github.com/frankaemika/libfranka

[2]https://github.com/ElementAI/MiniTouch

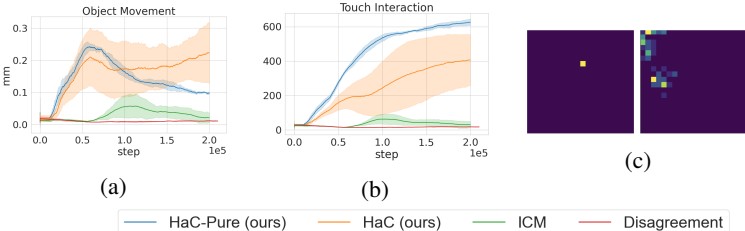

(a)  (b)  (c)

HaC-Pure (ours) — HaC (ours) — ICM — Disagreement

Figure 3: **Object Interaction** To quantitatively evaluate an agent's object interaction, we consider both haptics interaction and object displacement. (a) shows the object movement evaluation (avg displacement in mm from the starting point) of our method compared with the baseline methods over training steps; (b) depicts the number of touch interactions, evaluated on the "playing" task. (c) Heat maps showing the object location at the beginning (*left*) and end (*right*) of the HaC training. The object locations are spread-out towards the end indicating interactive movement.

- **Disagreement**: It uses model disagreement as objective for exploration [56, 28]. It leverages variance in the prediction of an ensemble of latent dynamics models as the reward.
- **RND**: Random Network Distillation [57] utilizes a randomly initialized neural network to specify an intrinsic reward for visiting unexplored states in hard exploration problems.

We built a Pytorch [58] version of these baselines based on their open source code (details in supplementary material). We use the following metrics to evaluate our method and the baseline models:

**Exploration success:** measures the percentage of times that the agent attained the goal state in the *exploratory* phase, i.e. with no external reward. Higher is better.
**Success:** denotes the percentage of times that the agent attained the goal state during the *downstream* task phase.
**Episode steps:** The number of steps required for each episode to succeed. This metric is an indicator of sample efficiency. The lesser the number of steps, the faster the agent's ability to succeed.
**Touch-interaction:** Amount of interaction the agent's fingers have with the underlying object. We measure this by computing the variance of force/torque sensory signal across the whole episode.
**Object movement:** The agent can resort to constantly engaging with the object unnecessarily to satisfy the objective. We, therefore, monitor the variance of door angle (for the Opening task) and the variance of object position (for the remaining tasks) over the course of training. A higher movement indicates diverse state space in addition to physical interaction.

## 5 Results and Discussion

HaC and baselines were trained on a Panda robot agent [54] for one million steps. In the *exploratory* phase of the training, we pre-train our method only with the curiosity-based intrinsic reward. We then progress to the *adaptation* phase. Also, note that across all tasks, HaC-Pure is based purely on cross-modal prediction, while HaC includes visual forward prediction reward in addition.

Figure 3a and 3b shows results on the basic task of playing with a single object. Since single object interaction does not have explicit goal states to evaluate, we instead measure the agent's ability to constantly engage and play with the underlying object. HaC-Pure displays four times better interaction with the underlying object when compared to SAC (see Figure 3b). Note from the plots that there is a trade-off when using HaC and HaC-Pure between constant interaction (i.e. touch interaction performance) and object movement dynamics. Collecting a variety of such interesting data during the *exploratory* phase helps the agent in terms of sample efficiency while solving the downstream tasks. Figure 8 shows similar comparison for Open-door task. In the following section, we examine the role of the forward objective term on touch interaction and possible ways to encode agent's observation in our ablation studies.

We compare HaC and HaC-Pure with SAC and state-of-the-art vision-based curiosity baselines on the remaining downstream tasks in MiniTouch. From Figure 4 it is evident that HaC and HaC-Pure perform better than SAC in all the tasks and better than the vision-based curiosity models in the majority of the tasks. Using SAC alone hinders the performance and is often unable to solve any of the three tasks. This is not surprising since the model is not motivated enough to collect diverse and useful data through interaction. We hypothesize that HaC-Pure without the visual

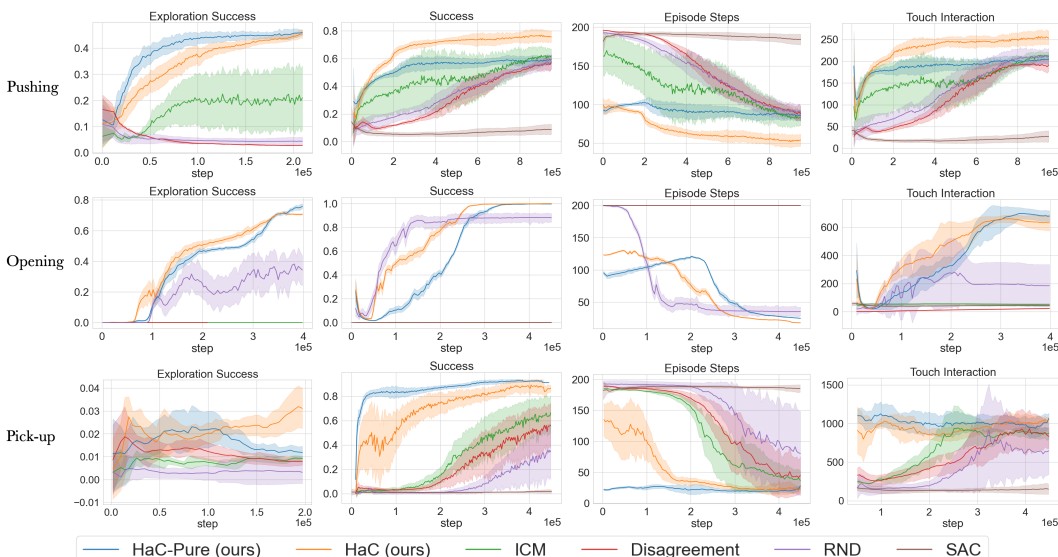

Figure 4: **MiniTouch Evaluations**. Each row in the figure outlines the performance of the HaC variants and the baselines on a MiniTouch task. Each column marks performance on the four specified evaluation metrics over a number of training steps on the x-axis expressed in $1e^5$. The results are averaged across 5 random seeds and shaded areas represent mean ± one standard deviation while darker line represents the mean. In the majority of the tasks, HaC agents attain success in the *exploratory* phase with no external reward (see text). *Note*: We exclude the single object playing task as success in this task is equivalent to object interaction as depicted in Figure. 3.

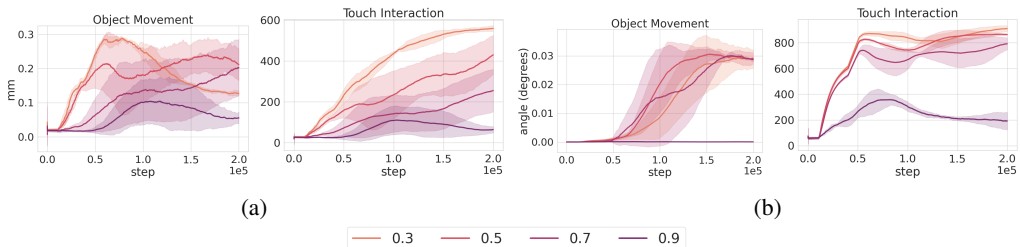

Figure 5: **Forward Prediction weight.** Touch and Object movement evaluated on the (a) playing task and (b) Open-door task for different forward prediction weightings sampled $\sim [1, 0]$. Large weight (darker plot) favors object-movement towards the end of training, where as smaller weight improves touch-interaction. Middle value in the sweep range (*e.g.* 0.5) balances the trade-off. Note that movement is measured as distance(mm) for the Playing and angular distance for Open-door.

prediction objective $L_{fdm}$ could potentially bias the solutions towards more physical interaction, which is not necessary for every task. For instance, constant interaction in the pushing task could be seen as a hindrance as it does not let the object slide easily towards the target. Observe that ICM performs better than HaC-Pure in the pushing task, however, HaC dominates in performance by about 15%. Recall that the goal is not just to succeed but to help attain success in a sample efficient manner in fewer steps. The results support our hypothesis that cross-modal curiosity enables an RL agent to succeed at an early stage in training and often without any external reward. Similarly, our model outperforms on the opening task without external reward. However, HaC initially has lower success compared to RND but surpasses RND towards the end. Although HaC and HaC-Pure attain similar success in the pick-up task towards the end of the training, it is compelling to note that HaC-Pure attains faster convergence. This is because the picking task requires constant touch interaction (where HaC-Pure has an advantage), as opposed to diverse object movement.

| Metric | Pushing | | | Open Door | | | Pick-up | | | Playing | | |
|---|---|---|---|---|---|---|---|---|---|---|---|---|
| | HaC | ICM+*haptics* | ICM | HaC | ICM+*haptics* | ICM | HaC | ICM+*haptics* | ICM | HaC | ICM+*haptics* | ICM |
| **Exploration** ↑ | **0.403** | 0.291 | 0.187 | **0.669** | 0.355 | 0.083 | **0.063** | 0.051 | 0.013 | - | - | - |
| **Success** ↑ | **0.733** | 0.678 | 0.597 | **0.983** | 0.571 | 0.114 | **0.891** | 0.825 | 0.780 | - | - | - |
| **Episode steps** ↓. | **57.84** | 87.61 | 95.24 | **23.34** | 97.10 | 199.3 | **30.54** | 33.77 | 42.19 | - | - | - |
| **Touch-interaction** ↑ | **247.79** | 210.11 | 202.66 | **600.1** | 287.97 | 43.56 | 980.7 | **984.2** | 952.3 | **388.15** | 267.021 | 63.31 |

Table 1: **Haptics-based future prediction** Table compares the mean evaluations for HaC and ICM+*haptics* on all the four tasks emphasizing the importance of cross-modal association(see text).
.

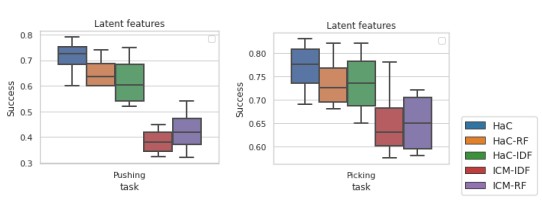

Figure 6: Comparing different latent space performance on *pushing* and *grasping* tasks. Features learned via haptics prediction perform better than those learned using IDF or Random.

## 5.1 Ablations

**Forward objective and tuning Lambda** Visual forward prediction $L_{fdm}$ plays an important role when it is used in the right proportion. Our intrinsic reward is a weighted combination of cross-modal prediction and forward prediction as defined in Eq.4. Figure 5 illustrates the model behavior with different levels of emphasis on the forward loss term, with $\lambda$ uniformly sampled between 0 and 1. Higher weights indicate that the future prediction dominates force/torque prediction. This leads to more object movement but lesser robot's constant touch-interaction with the underlying objects. This is useful in tasks such as Pushing. A smaller $\lambda$ value leads to better inactive behavior which is handy in tasks such as grasping. In our experiments we choose an intermediate value (*e.g.* 0.5) that works best for all of our tasks.

**Haptics in future prediction.** The goal of this experiment is to strengthen the argument of cross-modal association. While conducting experiments, it is important to deduce information of one modality from another modality in a related manner than simply adding another modality on top of visual information. We created an additional baseline, ICM+*haptics*, where in addition to the visual prediction model we include the haptics-based future prediction model. We hypothesize ICM+*haptics* to perform better than ICM as it has additional information (haptics). The haptics-based future prediction model takes a touch vector as input and predicts the touch vector for the next time step. Table 1 compares HaC and ICM+*haptics* on MiniTouch tasks and we observe that ICM+*haptics* is better than ICM but compares below HaC. Table 4 in the supplementary compares RND based baseline.

**Latent features for forward dynamics.** Choosing ideal embedding space for decoding the touch vector (Figure 1a) and for predicting future state (Figure. 1b) is important. Existing approaches rely on a pretext inverse dynamics (ID) task of predicting the agent's action given its current and next states [9]. Another simple yet strong method is to use features from a random but fixed initialization of the encoder [12]. In our work, we learn the features by leveraging the self-supervised pretext task of predicting one modality from the other. Figure 6 compares (1) encoder, i.e. learned through cross-modal prediction(HaC) (2) random feature encoder(HaC-RF) (3) encoder learned through ID task (HaC-IDF). In each case, the decoder network is optimized through touch prediction. We observe that the random features variant is stable and effective on both HaC and ICM models.

## 6 Conclusion

We formulated Haptics-based Curiosity (HaC) aimed at encouraging exploration via haptic interaction with the environment. We demonstrated in this work, that by involving additional modalities, the performance of curiosity-based systems on downstream tasks can be increased. We observed increased interaction with target objects, and presented evidence that HaC learns to solve the Mini-Touch benchmark tasks in an efficient manner while vanilla RL algorithms and vision-based curiosity formulations struggled. Force/torque sensing is widespread in the lab and industrial robots but while there are plenty of robotic benchmarks, we believe that tactile feedback is an under-explored modality and by releasing our benchmark, we hope to enable future research in this exciting area.

## Acknowledgements

We would like to sincerely thank Glen Berseth, Deepak Pathak, Edward Smith and Krishna Murthy for valuable feedback and insightful comments on the paper.

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
