# OpenReview forum: "Haptics-based Curiosity for Sparse-reward Tasks"
_robot-learning.org/CoRL/2021/Conference — CoRL2021 Poster_

### Official Review · Reviewer_vFAA · 2021-07-13

**Originality:** Good
**Technical Quality:** Fair
**Clarity Of Presentation:** Fair
**Impact:** 2

**Recommendation:**

Weak Accept: I recommend accepting the paper, but will not argue for my recommendation if the majority of other reviewers have a different opinion.

**Summary:**

This paper computes intrinsic motivation (i.e., artificial curiosity) by a vision and tactile sensing for improving the exploration in sparse reward function. Integrating tactile sensing is expected to guide the robot exploration in touch-intensive tasks better as the haptic information is more relevant to the rewards of these tasks. The proposed method trains an image encoder to transform an image input (x_t, where t denotes the timestep) to a latent vector z_t and then a decoder to reconstruct the haptic sensor inputs \hat{h}_t by the latent vector z_t. Furthermore, the author finds that incorporating forward dynamics model prediction objectives in vision inputs can improve exploration performance.


**Issues:**

1. What are the inputs of the SAC, ICM, Disagreement, and RND? The text seems that they only take images as inputs for both exploration and adaptation steps. If so, the comparison might be unfair. The author tries to argue that cross-modality (touch and vision) “association” is beneficial for exploration and, therefore, improves adaptation. To claim this, the author should compare with the baselines that take both vision and tactile sensing as inputs but don’t encourage the association between tactile and visual inputs. It is necessary to compare with these baselines because the performance gain of ToC could be from using multi-modality inputs.

2.  As the introduction said, you constrain the search space by touch because the task prior tells us that haptic novelty is more relevant to the tasks. That could be true in most touch-intensive tasks, but it might be misleading in some cases. For example, visually similar but materially different objects could constantly generate novelty to the robot and introduce noises in the reconstruction loss. I would like to see more discussion about this point.
I’m not sure of the meaning of the heatmaps in Figure 2. I can’t find the reference in the text, and the caption is confusing. Could you elaborate more on these heatmaps?

3. Did you train ICM, RND, and Disagreement with haptic features? That would be an important baseline for ToC. If ToC outperforms the baselines trained with haptic features or haptic + vision features, the importance of cross-modality association will be strengthened. The current experiments are not convincing to me that cross-modality association is the key for the performance gain in the experiments.

4. In the adaptation step, is each agent trained by images only or images+touch?

**Reviewer Expertise:**

Excellent: Expert knowledge on the topic of the paper

**Strengths And Weaknesses:**

Strengths:
- The idea of cross-modality association is still underexplored in curiosity-driven exploration.
- The author compares with plenty of baselines of curiosity-driven exploration algorithm.

Weakness:
- The experiment setup is unclear. For example, are the state inputs in both exploration and adaptation step the same? Do the baselines take touch and vision as state inputs?
- Some descriptions are vague. For example, L242,  "Figure 2a indicates that ToC is slightly more dynamic in the interaction
243 in contrast to ToC-Pure. ". What is "more dynamic"?
- Weak connection between the experiments and the proposed method. For example, wow does the proposed method ToC improve generalization to novel objects? Generalization is important but it seems to me ToC is not expected to improve generalization over novel objects. Also, it would be better to see the generalization ability of other methods.

**Summary Of Recommendation:**

Overall, the idea is well-motivated, and the cross-modality curiosity is promising. Most curiosity algorithms only use single modality information. I think this paper did a good job of bringing up the attention on cross-modality association in curiosity-driven algorithms. That being said, the reasoning for the experimental results is unclear to me in the current version of the manuscript. I would like to adjust my rating after rebuttal.

---

> ### Author Response · Authors · 2021-08-27
> **Response to Reviewer vFAA**
>
> We’d like to thank the reviewer for the insightful comments and their time spent reviewing our work!
>
> **“What are the inputs of the SAC, ICM, Disagreement, and RND?”**
>
> SAC, ICM, Disagreement, and RND take image-based inputs. In our approach, to be able to compare with the baselines, our agent uses image features as inputs and does not have access to touch information. We compare with the baselines that take both vision and tactile sensing as inputs but don’t encourage the association between tactile and visual inputs, as detailed in the next response.
> On a high level, we want to show that leveraging another modality does improve the performance (comparing with the case when you don’t). Especially in continuous control tasks, we encourage to use touch sensory information, if available, in a manner that would improve underlying performance on the downstream tasks. Thanks for pointing out that this was confusing. We have updated the experimental setup section for clarity.
>
> **Regarding training touch+vision baselines and comparisons:**
>
> Thanks, this is a great point! To show that cross-modal consistency is useful, we have added an additional prediction-based baseline (Toc-future) in Table. 1 which is essentially the ICM version of touch+vision. In the revised version, we called this ICM+Touch to be more explicit. The prediction space is a concatenated vector of touch and visual features. The method predicts the touch+visual feature of the next time-step from the current input. We observe that next state prediction approaches are effective in cases where object movement is essential but underperforms when it comes to constant interactive exploration with the underlying objects (e.g. opening, picking). We have also included a similar comparison table with an additional baseline in the revised version of the paper (Table. 2). It is based on random network distillation using vision and touch. Here we again used the touch+visual feature vector (in a concatenated fashion) as input to both the random target network and the predictor network.
>
> **Regarding “Some descriptions are vague, What is "more dynamic"?”**
>
> - One issue with just using touch-based curiosity is that it could lead to unnecessary physical interaction (touching) to satisfy curiosity objective.
> - This could result in reduced object movement.
> - This might hurt performance for some downstream tasks (Eg: pushing)
> - For the Pushing task, the agent could either interact dynamically with high directed velocity to push the object towards the target, or the agent could resort to touching it more often and making slow progress towards the target.
> - Touch-curiosity reward along with Forward prediction rewards (ToC) inherently balances this trade-off.
>
> We updated the text to make the context more clear.
>
> **Regarding heatmaps in figure 2**
>
> The two heatmaps in Figure. 2(c) essentially depict object movement at the beginning of training and towards the end of training respectively for the playing task. Specifically, positions on the screen where the object is located. In the heat map on the right of Figure. 2(c) the object location is mostly focused at a single position while on the right the plausible locations on-screen are spread out over the episode indicating that it is being moved often by the arm towards the end of training.
> Similarly, the two heatmaps in Figure. 2(d) shows the possible arm positions in the beginning and towards the end of the training. Since both were correlated, we retained just one of them in the revised version with the appropriate description. Please note that Figure. 2 is now Figure 3 in the revised version.
>
> **Regarding “weak connection between the experiments and the proposed method”**
>
> Generalization to novel shapes is certainly not the mainstream problem we are tackling and is not composed of our main experimental results. We added this proof of concept results in our ablations to investigate that the model parameters in the exploratory phase are not just reusable across different tasks with similar shapes (Eg: Playing, Pushing, and Picking), but also can help generalize when applied to distinctive shapes. In the revised version, we conducted additional experiments where we compare this behavior across different exploration techniques in the additional ablation section (Table 3). Our method's performance  holds promise for future work exploring the hypothesis that - the behavior learned via multi-modal consistency can serve as a better initialization to learn repurposable features across different tasks and with distinctive objects.
>
> **Regarding “In the adaptation step, is each agent trained by images only or images+touch?”**
>
> The downstream adaptation phases use image input as mentioned. We use touch only in the exploration phase. We have updated the text to make this clearer.

---

> > ### Comment · Reviewer_vFAA · 2021-09-02
> > **Follow-up**
> >
> > Thanks for the response. The response addressed my several concern. I appreciate the author added "ICM+Touch" baseline.

---

> > > ### Author Response · Authors · 2021-09-02
> > > **Response to follow-up**
> > >
> > > We are really glad to hear the concerns have been addressed. We have also added the "RND+Touch" baseline in the revised appendix  (Table 2). Once again thank you for the comments which helped us improve the paper!

---

### Official Review · Reviewer_6jo7 · 2021-07-22

**Originality:** Fair
**Technical Quality:** Good
**Clarity Of Presentation:** Good
**Impact:** 3

**Recommendation:**

Weak Accept: I recommend accepting the paper, but will not argue for my recommendation if the majority of other reviewers have a different opinion.

**Summary:**

This paper presents a curiosity-based approach that leverages both vision and "touch" as a self-supervised pretext task. The task is to predict the observed haptics signal from the visual input (Fig 2).  Experiments show the approach leads to better performance on downstream sparse-reward RL tasks. The authors also contribute a benchmark (minitouch) can that be used with their method to pre-train networks and doesn't require a proprietary simulator.

**Issues:**

See above for specific issues.

**Reviewer Expertise:**

Very good: Comprehensive knowledge of the area

**Strengths And Weaknesses:**

### Major comments
In general, this paper is clear and easy-to-follow. The key idea (cross-modal learning) is simple and easy to implement, and the experiments validate the approach to a good degree (up to the points below).

As for weaknesses, the technical advance appears to be incremental given that previous work (e.g., those in Sec 2 and [47]) have shown the benefits of cross-modal learning for learning better representations. The paper also has many minor errors that should be corrected before it can be accepted. Please see below for specific questions/concerns:

- line 147: Eq (2) and statement that "A high prediction error on a given image indicates that the agent has had few interactions like this." This is not necessarily true as the image may not be informative of the tactile sensation, e.g., consider real textured surfaces v.s. a flat printed texture. Essentially, the underlying predictive distribution may be multi-modal (as in multiple peaks and not multiple sensors) and the prediction error may not be informative.
- Can the authors confirm that the touch information is sparse, i.e., only occurs when there is contact? If so, then the decoder would have zero output for a large amount of the state space when there is nothing in contact with the arm?
- In the experiments and Fig 5(b), the authors claim "robustness to diverse shapes": how different are the unseen objects? The performance variance on the unseen tasks appears larger.
- A short discussion of how the approach can be used in the real-world would help to increase the paper's impact. From the experiments, it seems a large number of steps are needed (20k)? Also, it's unclear if each "training step" is an episode or a time-step? The paper also mentions "episode steps" and it is unclear what is an episode is in the setup.
- Fig 2 (c) and (d) are unclear. Are the x,y-axis coordinates? which coordinate frame? I don't understand what is meant by "Robot’s ﬁngers are spread everywhere at starting while learn to focus near the object towards the end of training." Also, where is the object? at the top left?
- Fig 3: the pushing task in the "Success" "Episode steps" and "Touch interaction"  seem to be over 1e6 steps rather than 1e5?
- Fig 3. The episode steps appear to start at a different number of steps for ToC and ToC-pure? Are these episode steps collected from the adaptation phase?

### Minor issues:
- Tuning $\lambda$ for the forward objective weight may be an issue and some discussion on how to pick it would help.
- There is related work in learning tactile skills through exploration (Pape et al, 2012) and multimodal learning (Chen et al, 2021) that attempts to learn consistent representations from multiple modalities for RL.
- 188: "two-layerd"->"two-layered"
- 214: "Network Distiallation" -> "Network Distillation"
- Fig 3: "darker line represents mean" -> "darker line represents the mean"
- Table 1 caption: "Toc-Pure" is "ToC-future"?
- 257: "to succeeds" -> "to succeed"
- 259: "surpasses towards" -> "surpasses RND towards"
- 295: "figure.5a" -> "Figure 5a"
- pg. 5 . Two footnotes with the same number (1).

### References

- Pape et al, 2012, Learning tactile skills through curious exploration, Frontiers on Neurorobotics.
- Chen et al, 2021, Multi-Modal Mutual Information (MuMMI) Training for Robust Self-Supervised Deep Reinforcement Learning, ICRA 2021.


**Summary Of Recommendation:**

Overall, I like the idea of doing cross-modal learning. However, the paper is borderline and falls slightly short due to the limited technical novelty and the numerous minor errors.

### Post Response
Thank you for responding to my comments. I have amended my recommendation accordingly.

---

> ### Author Response · Authors · 2021-08-27
> **Response to Reviewer 6jo7**
>
> Thanks for your valuable comments.
> We particularly thank the reviewer for all the corrections in the “minor issues” section and we’ve updated the paper accordingly. Also would like to thank you for pointing out Pape et al and Chen et al, which we added to our references.
>
> **Regarding "technical advance appears to be incremental given that previous work"**
>
> We agree that many multi-modal-based papers investigate the usefulness of incorporating additional modalities. However, we would like to clarify that although related, we are less interested in learning representations and rather emphasize more learning interesting exploratory policies using cross-modal consistencies.
> Although an incremental advancement, our technique seems to work and can be considered a useful tool for any continuous control tasks in which task-unrelated data can be obtained at low cost. We observed that most RL curiosity-based methods failed on those tasks where interactive manipulation of objects is needed. In our case, the agent is encouraged to learn relevant exploratory behaviors autonomously in the initial phase and bootstrapping the attained self-play knowledge in downstream tasks. We also demonstrate in our experiments that behavior that is learned in this way task-independently can be reused in multiple downstream tasks.
>
> **Regarding "image may not be informative of the tactile sensation, e.g., consider real textured surfaces v.s. a flat printed texture"**
>
> You are right that in this method’s current form we make the assumption that objects are somewhat homogenously textured and for example not covered in a checkerboard pattern. One straightforward way to address this limitation is to incorporate a generative model like a VAE instead of our encoder-decoder framework to learn visual priors that help decouple objecthood from visual presentation.
>
> **Regarding how the approach can be used in the real-world would help to increase the paper's impact. From the experiments, it seems a large number of steps are needed (20k)?**
>
> The reviewer is bringing up a good point. Our general reasoning here is that it’s possible to define safety boundaries for a robot arm (e.g. don’t hit the table, don’t move outside the arena boundaries) and have a robot arm autonomously and task-independently collect data for phase 1 of our method. In addition to this, we would like to point out that (a) in the pushing and pick-up tasks, only 5-10k steps are required for decent performance on the downstream task and (b) what we call a “step” is however long it takes the inverse kinematic solver to move the arm to the new end effector pose (that was generated by the policy). If many of the generated poses cluster together, that dramatically reduces runtime of the curiosity phase. But the reviewer is correct that this needs to be addressed in the paper and we have added a section discussing this to the appendix.
>
> **Regarding “it's unclear if each "training step" is an episode or a time-step?**
>
> Each training step here is a time step.
>
> **Regarding “Can the authors confirm that the touch information is sparse?**
>
> *Response:* Touch information is indeed sparse in general. However, we're applying this to contact-rich tasks like pushing an object where sparsity occurs in a controlled manner. We also tried GAN like discriminator model to learn the cross-modal associations and it didn't work as using a discriminator leads to highly ambiguous outcomes in a sparse setting.
>
>
> **Regarding Fig 2 (c) and (d) :**
>
> *Response:* Figure. 2(C) is the heatmap of probable object locations in 2D coordinates for the playing task at the beginning and end of exploratory training. In the beginning, since the object is not moved by the arm the object location is static at a single position (yellow being the most probable location and dark blue being the least probable location). Towards the end, the positions are a bit more spread out indicating increased object movement (mostly due to interaction). On the other hand Figure. 2(d) shows the heatmap of the possible location of the robot arm at the beginning and end of the training. In the beginning, the arm position is quite random and is almost everywhere, while towards the end it converges to a smaller area (mostly focused around the object locations) indicating increased interaction. The locations are as seen from the camera of the robot. Please note that Figure. 2 is now Figure 3 in the revised version.
>
> **Regarding “the pushing task in the "Success" "Episode steps" and "Touch interaction" seem to be over 1e6 steps rather than 1e5?”**
>
> *Response:* Sorry for the confusion here, this is because the values on the x-axis for the pushing task are already at a fraction lower (Eg: 0.2e6 instead of 2e5). This has been fixed in the revised version.

---

> > ### Comment · Reviewer_6jo7 · 2021-09-01
> > **Thanks!**
> >
> > Thanks for responding to my comments and addressing the errors. I have read through the response and other reviews. I like the general direction of this work and I do think that cross-modal curiosity (esp related to haptics) is an interesting avenue to explore. I have raised my recommendation to a borderline accept.

---

> > > ### Author Response · Authors · 2021-09-02
> > > **Thank you!**
> > >
> > > We are glad our response has addressed the comments and we thank the reviewer for supporting our paper!

---

### Official Review · Reviewer_2Equ · 2021-07-26

**Originality:** Good
**Technical Quality:** Good
**Clarity Of Presentation:** Good
**Impact:** 4

**Recommendation:**

Weak Accept: I recommend accepting the paper, but will not argue for my recommendation if the majority of other reviewers have a different opinion.

**Summary:**

The work presents an approach that considers visuo-tactile representations for addressing tasks such as pushing objects and opening door, etc. (in simulation). In the proposed method, cross-modal consistency between visual and touch signals is used to guide exploration in reinforcement learning tasks, i.e., mismatches between visual features and touch representations will lead to surprises and more exploration. Two variants of the method are studied: Touch-based Curiosity (ToC) and Pure ToC, i.e., with the intrinsic reward being derived both from the visuo-tactile representations and vision-based predictive model, and visuo-tactile representations only. The experiments show that this approach can obtain better results than the standard (vision-based) Soft Actor-Critic algorithm and the state-of-the-art vision-based curiosity method ICM (SAC +  Intrinsic Curiosity Module).

**Issues:**

1. Title: Touch-based isn't entirely correct as the work doesn't consider any touch sensors per se and rather a force/torque vector.
2. A few settings of the methods are not clear. Why are the touch features only used to craft the intrinsic rewards? Why does the input to RL agent consists of visual features alone? If both touch features and visual features are taken as the input to the RL agent, the performance of the RL agent may be better. As for the loss functions, why was the L2 reconstruction loss selected instead of L1 loss?
3. Is there a better way to  get optimal lamda (Eq. 4) than uniformly sampling it between 0 and 1? Also, it may not be optimal to use a fixed lamda, has a changeable been attempted to maximize the performance?
4. Introduction / Paragraph 2: It would be interesting to find an argument for exploration from a computation standpoint. What's the practical advantage of animals and infants being curious? Why is it preferable to random exploration?
5. Figure 1:  the representation of the loss function should be consistent between figure (a) and (b), to ease the paper reading.
6. In the caption of Figure 1, "actual next latent state zˆ_t" should be "actual next latent state z_{t+1}".
7. Related Work / Paragraph 4 (101-107) should be revised, other works have studied visuo-tactile representations.
Lee, J.T., Bollegala, D. and Luo, S., 2019, May. “Touching to see” and “seeing to feel”: Robotic cross-modal sensory data generation for visual-tactile perception. In 2019 International Conference on Robotics and Automation (ICRA) (pp. 4276-4282). IEEE.
8. Related Work / Paragraph 5 (114-115) "This might not work in our case as touch is a more sparse signal and using a discriminator could lead to ambiguous outcomes." This sentence seems to be ambiguous, as it is dependent on the used sensor, and not backed by any evidence.
9. Problem Formulation (132) is not correct, pi_star is the optimal policy that maximizes the discounted sum of rewards.
10. Figure 2: It is not clear what (c) and (d) mean.
11. Figure 4: Which one is better? More touch information and more object movement? Why? Why are there 4 plots (either angle and mm)?
12. Figure 5(b) shows good generalization to novel objects, but there is no details about the seen dataset and unseen dataset.
13. Table 1: The column refers to the method as ToC and ToC-fut, the caption as ToC and ToC Pure. In other parts of the paper, the same ambiguity persists e.g. 274-281.
14. Lines: 290-298: What are random features? (e.g. reference the section) and what are IDF?
15. Figure 5: It would be better to add extra spacing between figures (a) and (b) to help the interpretation, and use one single legend in figure (b)  to be consistent with figure (a).
16. Some grammar issues:
line 118: a RL agent -> an RL agent
 line 145: a L2 -> an L2

**Reviewer Expertise:**

Excellent: Expert knowledge on the topic of the paper

**Strengths And Weaknesses:**

Strengths:
1. The idea of cross-modal curiosity for reinforcement learning tasks is interesting.
2. The experiments show that this approach can obtain better results than the standard (vision-based) Soft Actor-Critic algorithm and the state-of-the-art vision-based curiosity method ICM (SAC +  Intrinsic Curiosity Module).

Weaknesses:
1. Only simulation experiments are presented and no real robot data have been used in the experiments.
2. The presentation of the paper can be improved.

**Summary Of Recommendation:**

The work presents an interesting idea of cross-modal curiosity for reinforcement learning tasks but it lacks proper justification of the motivation of cross-modal curiosity and needs further validation on the real robots.

---

> ### Author Response · Authors · 2021-08-27
> **Response to Reviewer 2Equ**
>
> We appreciate the reviewer’s positive comments and valuable feedback. Here are our responses regarding the issues above.
>
> **Regarding the title:**
>
> Force-Torque sensors are more common in robotics. We understand that dedicated touch sensors that output high-resolution touch information are not uncommon anymore. Although our method can be potentially applicable to other forms of touch, we do not want to appear to be biting off more than what we could chew. That said, we are happy to change the title to make the content more explicit, stating the fact that we use Force/Torque vector in this work. If the reviewer has any ideas on what a more suitable title could be, we’d be open to suggestions. For now, we are thinking “Haptic Curiosity..” or “Hand-based Curiosity..” but we are afraid that both also come with their share of non-force/torque associations. And “Force-Torque Curiosity…” doesn’t quite roll off the tongue.
>
> **Using the touch for the intrinsic rewards alone:**
>
> We designed our approach to make it easier and fair compared to vision-only baselines. We made sure that our RL agent has access only to the visual features. As a result, the touch features are only used in learning regularities by providing a strong intrinsic reward for exploring novel state space.
>
> **What's the practical advantage of animals and infants being curious? Why is it preferable to random exploration?**
>
> We thank the reviewer for this interesting question. Much of the work of Eleanor [1] and James [2] Gibson revolved around this same idea. We believe one of the main findings in their work was that action shapes perception, i.e. that in mammals, visual learning is significantly easier if the visual stimuli can be associated with haptic or other sensations. Other cognitive scientists like Piaget argue that this could be due to a preference for causal association [3], i.e. associating multiple modalities (smell, touch, taste) and physical properties (obtained from interaction) with an object can help determine the uses for this object.
> Computationally, we would argue, that any task pressure improves representation learning for that task. For example, if the downstream tasks all involve manipulation, then anything that forces your representation to encode manipulation-related information will lead to greater performance on the downstream task.
>
> **Regarding Lambda**
>
> For each of our tasks, we found that a value that is intermediate in the range [0.4, 0.6] has been optimal for most of the tasks(Eg: avoiding unnecessary static interaction). Rather than using different Lambda for each of the tasks, we, however, chose the value 0.5 and fixed it to make the model straightforward and reproducible.
>
> **Regarding Figures**
>
> We added the suggested spacing and legend for Figure.5. We updated the caption for Table.1 and also corrected mentioned grammatical issues.
>
> **heatmaps in Figure2**
>
> The two heatmaps in Figure.2 (c) (now updated to Figure.3(c))  depict positions on the screen where the object is located. In the heat map on the right of Figure.2(c) the object location is mostly focused at a single position while on the right the plausible locations on-screen are spread out over the episode indicating that it is being moved often by the arm towards the end of training.
>
> **Fig4; Why are there 4 plots (either angle and mm)?**
>
> Figure.4 includes touch interaction and movement plots for both pushing and opening tasks respectively. We used distance metric (mm) for the pushing task and angular displacement for the door opening task. We made subfigures for each of the tasks in Figure.4 (which is now Figure.5) for more clarity.
>
> **This sentence seems to be ambiguous, as it is dependent on the used sensor**
>
> We agree that the sensor apparatus needs to be considered for a more accurate claim. However, compared to vision and audio where the information is continuous, touch-based input signals, in general, are locally continuous and sparse in nature( i.e.,only occurs when there is contact). We rephrased the sentence to make it clear that this is especially true in our setting with Force/Torque vectors.
>
> **What are random features? and what are IDF?**
>
> IDF stands for features that are learned using Inverse Dynamics (ID) prediction task. Recent works showed that the features from randomly initialized encoder work suitably well for large-scale settings[4]. We term the Random Features here as (RF). We added references accordingly and made the abbreviations clear in the revised version.
>
> We have included the citation for Lee et al. Thank you for the reference.
>
> - [1] Gibson, E. J., & Walker, A. S. (1984). Development of knowledge of visual-tactual affordances of substance.
> - [2] Gibson, J. J. (2014). The ecological approach to visual perception: classic edition. Psychology Press.
> - [3] Piaget, J. (1977). The role of action in the development of thinking. In Knowledge and development.
> - [4] (2018) Burda.Y. Large-Scale Study of Curiosity-Driven Learning.

---

> > ### Comment · Reviewer_2Equ · 2021-09-03
> > **Thanks for your response and interesting discussions**
> >
> > Thanks for responding to my comments and the nice discussions replying to my questions. I appreciate the effort and time the authors have put into addressing the questions raised in the reviews.
> >
> > As for the title, I believe "Haptic Curiosity" would be appropriate as "haptic" can be referred to as the force/torque measurements for single point contacts in a narrow sense and be referred to measurements related to the sense of touch that includes forces/torques in a broad sense.
> >
> > I would like to maintain my positive rating.

---

> > > ### Author Response · Authors · 2021-09-04
> > > **Thank you**
> > >
> > > We appreciate the reviewer's suggestion regarding the title. We will update the title accordingly in the accepted version. Many thanks for your effort in improving our paper and positive rating.

---

### Official Review · Reviewer_5q2G · 2021-07-27

**Originality:** Good
**Technical Quality:** Very Good
**Clarity Of Presentation:** Very Good
**Impact:** 3

**Recommendation:**

Weak Accept: I recommend accepting the paper, but will not argue for my recommendation if the majority of other reviewers have a different opinion.

**Summary:**

This paper develops a touch-based curiosity approach for touch-intensive robot manipulation tasks with sparse rewards. The paper describes a two-stage learning approach where in the first stage, exploration is performed based on cross-modal (visual and touch) consistency and mismatch that inherently encourages robot-object physical interactions and in the second stage, the parameters from the exploration stage are re-used to achieve a downstream task with a task-specific sparse reward.


**Issues:**

See “Suggestions for Improvement” section above.


**Reviewer Expertise:**

Excellent: Expert knowledge on the topic of the paper

**Strengths And Weaknesses:**

Strengths:
- Exploration based on cross-modal consistency that encourages physical interaction based on haptic modality mismatch and augmenting it with visual surprise based on a forward dynamics model
- Consideration of a variety of metrics including object movement to avoid passive physical interactions
- A benchmark with four simulated manipulation tasks with F/T sensor data: playing, pushing, opening, and picking up
- Baseline comparison with single-modality approach

Questions / Suggestions for Improvement:

- In haptics, sensing inherently depends on action. What this means is, forces during a physical interaction are not only dependent on object states (object physical characteristics) but also on robot states (what was the impact velocity etc.). The touch-control module, which maps visual signals to haptic signals for checking haptic modality mismatch does not consider robot states explicitly for this mapping. Can the paper comment on how will the cross-modal consistency generalize across different robot control behaviors (for example: different robot velocities for pushing etc.)?

- What happens if the task success reward during the second phase is actually simultaneously used with the touch interaction reward in a single stage itself? Can a comparison be done with the two-stage approach?

- Can the parameters from the exploratory phase be reused across multiple tasks if all it’s getting is touch experience from visual inputs? Insights would be useful. For example, exploratory tasks would just focus on playing with the object and learning more about the visual-haptic mapping. And then, you use that to perform various tasks with the object such as pushing, opening, pick-and-place etc.?

- Can the paper discuss how the methods would deal with physical interactions that may require repeated make and break of contacts, such as in some in-hand manipulation scenarios?

- In a related way, do the methods bias the solutions towards more physical interaction even if they are not necessary? For example, a pushing task can also be achieved by increasing the robot’s impact velocity and letting the object slide on a table-top once the model has a good understanding of the physics. Can the paper discuss this and provide some insights?

- Some more insights about the results would make the paper stronger such as: 1) Why is TOC-future worse than TOC? 2) Why does the visual-curiosity based approach (ICM) perform better than TOC-pure (touch-based curiosity) for the pushing task? Is it because for the pushing task, object movement could probably be more crucial?

- “Each of the fingers is equipped with a simulated force/torque sensor that measures the joint reaction force applied to it” - ? Why is there a 6D F/T sensor on each finger instead of one at the wrist or a simple tactile sensor on each finger?

- Fig.1 - caption: \hat{z_t} -> \hat{z^_{t+1}}

- It would be very helpful to the readers if the graphs are made more legible with bigger fonts, larger frame sizes etc.


**Summary Of Recommendation:**

The paper describes an interesting two-stage learning approach which encourages physical interaction by considering cross-modal consistency and forward dynamics in the exploratory stage and sparse task rewards in the adaptation stage. As long as the limitations of the approach are clearly identified and the points raised above are discussed in the revised manuscript, I would be in favor of publication of this manuscript.

---

> ### Author Response · Authors · 2021-08-27
> **Response to Reviewer 5q2G**
>
> We appreciate the reviewer’s feedback and valuable suggestions. Here are our responses regarding the reviewer’s questions.
>
>  **Regarding cross-modal consistency behavior across different robot control behaviors:**
>
> *Response:* Force/Torque values in a manipulation task are indeed a function of both object characteristics and robot state. We added a discussion in the paper that summarizes the behavior for this setting. We observed that the impact of our method is lower when the object mass is too low or when the robot forces are substantially lower, in this case increasing the weight on cross-modal consistency term as opposed to future prediction term of the ToC objective is helpful. On the contrary, when the object mass and robot forces are large it hurts the task performance as the agent is happy interacting with the underlying object, as it easily obtains larger intrinsic rewards. To discourage such behavior we could use a larger weighting on the forward-dynamics objective.
>
> **Regarding “What happens if the task success reward during the second phase is actually simultaneously used with the touch interaction reward in a single stage itself?”**
>
> *Response:* We (deliberately) didn’t run this yet. “Deliberately” because we wanted a clean separation between task-independent exploration and downstream task.  We also wanted to have a fair comparison with baselines that take visual features as input during the task phase. However, the reviewer is making a good point that this would be interesting. We can’t promise that we’ll have time to run this before the decision deadline but we’re trying to include these experiments in the camera-ready version of the paper.
>
> **Regarding “Do the methods bias the solutions towards more physical interaction even if they are not necessary?”**
>
> *Response:* We hypothesize this could be an undesired outcome of our ToC-Pure technique (Eg: when the object mass (eg: door mass) is too large). Precisely to avoid these undesired effects, our full model includes a forward regularizer term that discourages static interactive behavior. We believe that a moderately tuned ToC approach can yield both the benefits for an interactive exploratory policy i.e. having both desired interaction and dynamic object movement.
>
> **Regarding “Can the parameters from the exploratory phase be reused across multiple tasks if all it’s getting is touch experience from visual inputs?”**
>
> *Response:* This is close to what we do in our approach, we train on the exploratory stage (which is shared between the Pushing, Playing, and Pick-up tasks) and then transfer the learned behavior to the task phase. This can also be seen as one of the advantages of modular design choice of separating exploration phase from downstream task phase. In addition, we show evidence of this outcome in our ablations. We investigate the generalization ability to novel objects in Figure. 6(b). We believe that this could also be an independent large-scale study by itself. In the follow-up work, we are further exploring the approach in a larger Meta-RL framework. We hypothesize that the exploratory behavior learned via multi-modal consistency can serve as a better initialization to learn repurposable features across different tasks.
>
> **Regarding “Why does the visual-curiosity based approach (ICM) perform better than TOC-pure (touch-based curiosity) for the pushing task?”**
>
> *Response:* The reason is as anticipated, better performance on the Pushing tasks would require both interaction and active object movement. Constant engagement with the arm is not very crucial for the pushing tasks as much as for the Pickup task. We made sure to add the relevant explanation in the revised version. On the contrary, ToC-future (which we renamed as ICM+touch to avoid ambiguity) shines in the places where dynamic movement is required but doesn’t quite encourage interaction as ToC.
>
> ---
> We thank the reviewer for the formatting remarks. We have corrected the Figure. 1 caption and also tried making graphs slightly larger (this is due to space constraints and page limit)

---

### Meta-Review · Area_Chair_5ET2 · 2021-08-17

**Recommendation:** Accept (Poster)
**Confidence:** 4

**Metareview:**

This paper presents a multi-modal algorithm to improve exploration in presence of torque/force readings

Quality:

(+) The experiments seem to support the hypothesis of the manuscript

(-) Reviewer vFAA highlighted several clarifications and additional experiments that would be nice to perform.

(-) Experiments are only performed in simulation

Clarity:

(+) The paper is overall well written

(-) All reviewers suggested several changes to the text to make it more accurate and clear.

(-) As remarked by Reviewer 2Equ the paper does not indeed make use of touch sensors, but only of torque/force sensors. The  title and a large part of the nomenclature in the paper is thus misleading and should be modified to accurately reflect the content of the paper

Originality:

(+) The mixture of curiosity-based systems and multi-modal sensing (vision and force/torque) is novel

Significance:

(+) The reviewers agree that the topic tackled in this manuscript is interesting and potentially impactful.

(-) Reviewer 6jo7 had concerns regarding the significance of the contribution compared to existing work


In summary, while the paper deal with an interesting topic, the current manuscript seems to have several limitations, from experiments that should be improved, to clarity, and overall contribution of the paper.


Minor comment:
Recently, several simulators for tactile sensors have been open-sourced (e.g., https://github.com/facebookresearch/tacto, https://github.com/robotology/gazebo-yarp-plugins/issues/55) why not using one of them instead of force/torque sensors?

---

After rebuttal:
The authors improved the paper and addressed most of the concerns from the reviewers.
There now seems to be agreement from the reviewers that the paper is sufficiently solid.
Although the title has not been changed in the manuscript, I do expect the authors to change it as agreed in the rebuttal, and remove all mentions of "touch sensing" from the manuscript.

Minor comment: Fig 4, 5, 6 are hardly readable, and completely unreadable when printed in B&W. You should consider making the figures larger and more readable.

---

> ### Author Response · Authors · 2021-08-27
> **Response to Area Chair 5ET2**
>
> We thank the Area Chair for the review summary. We appreciate the valuable feedback. We would like to address the points here:
>
> **Regarding experiments:**
>
> We addressed the clarification raised by Reviewer vFAA, and added additional experiments that compare our technique with baseline+touch. We already have an experiment showing the comparisons with ICM+touch baseline, and in the revised paper we extended the results setup to the RND baseline in Table. 2. We have also included Figure. 10 in the revised version showing additional experimental comparisons with the baselines conducted on generalization behaviour.
>
> **Regarding Clarity:**
>
> We incorporated the reviewers’ suggestions and desired changes into the text. We acknowledge reviewer 2Equ’s comment on the title of the paper and we are happy to change the title in the accepted version to be more explicit and clear that we are using F/T sensors in our work. Thanks to the open discussion format here at OpenReview, we would like to brainstorm some possible options with the reviewer.
>
> **Regarding Originality:**
>
> We tried to address the concerns of Reviewer 6jo7 in our response.
>
> **Regarding the new tactile simulators:**
>
> Thanks for pointing this out! We weren’t aware of the TACTO work. The reason we opted for F/T sensing, in general, is that we believe these are ubiquitous on robot arms in research (or at least significantly more common than vision-based haptic sensors).
>
>
> ### Summary of changes
>
> Based on the reviews, we have incorporated the following revisions to our submission:
>
> - We included missing citations suggested by the reviewers in the literature section.
> - Added the experiments and comparisons suggested by Reviewer vFAA.
> - We improved the writing and corrected grammatical errors in the text as suggested by the reviewers.
> - We are happy to change the title to be more explicit that we leverage F/T sensors in our work.
> - Added a discussion in the supplementary material on how our technique can be used in the real world.
>
> We thank again all the reviewers and the area chair for proving constructive comments to improve our work. We have also provided individual responses for each reviewer to address the issues and concerns.

---

### Decision · Program_Chairs · 2021-09-13

**Decision:**

Accept (Poster)

**Comment:**

This paper presents a multi-modal algorithm to improve exploration in presence of torque/force readings

Quality:

(+) The experiments seem to support the hypothesis of the manuscript

(-) Reviewer vFAA highlighted several clarifications and additional experiments that would be nice to perform.

(-) Experiments are only performed in simulation

Clarity:

(+) The paper is overall well written

(-) All reviewers suggested several changes to the text to make it more accurate and clear.

(-) As remarked by Reviewer 2Equ the paper does not indeed make use of touch sensors, but only of torque/force sensors. The  title and a large part of the nomenclature in the paper is thus misleading and should be modified to accurately reflect the content of the paper

Originality:

(+) The mixture of curiosity-based systems and multi-modal sensing (vision and force/torque) is novel

Significance:

(+) The reviewers agree that the topic tackled in this manuscript is interesting and potentially impactful.

(-) Reviewer 6jo7 had concerns regarding the significance of the contribution compared to existing work


In summary, while the paper deal with an interesting topic, the current manuscript seems to have several limitations, from experiments that should be improved, to clarity, and overall contribution of the paper.


Minor comment:
Recently, several simulators for tactile sensors have been open-sourced (e.g., https://github.com/facebookresearch/tacto, https://github.com/robotology/gazebo-yarp-plugins/issues/55) why not using one of them instead of force/torque sensors?

---

After rebuttal:
The authors improved the paper and addressed most of the concerns from the reviewers.
There now seems to be agreement from the reviewers that the paper is sufficiently solid.
Although the title has not been changed in the manuscript, I do expect the authors to change it as agreed in the rebuttal, and remove all mentions of "touch sensing" from the manuscript.

Minor comment: Fig 4, 5, 6 are hardly readable, and completely unreadable when printed in B&W. You should consider making the figures larger and more readable.